# *UMOD* Polymorphisms Associated with Kidney Function, Serum Uromodulin and Risk of Mortality among Patients with Chronic Kidney Disease, Results from the C-STRIDE Study

**DOI:** 10.3390/genes12111687

**Published:** 2021-10-23

**Authors:** Jinwei Wang, Lili Liu, Kevin He, Bixia Gao, Fang Wang, Minghui Zhao, Luxia Zhang

**Affiliations:** 1Renal Division, Department of Medicine, Peking University First Hospital, Beijing 100034, China; gslzwjw@163.com (J.W.); cherrylee_liu@163.com (L.L.); sherrygao1021@126.com (B.G.); wangfang@bjmu.edu.cn (F.W.); mhzhao@bjmu.edu.cn (M.Z.); 2Peking University Institute of Nephrology, Beijing 100034, China; 3Key Laboratory of Renal Disease, National Health Commission of China, Beijing 100034, China; 4Key Laboratory of Chronic Kidney Disease Prevention and Treatment, Ministry of Education of China, Beijing 100034, China; 5Research Units of Diagnosis and Treatment of Immune-mediated Kidney Diseases, Chinese Academy of Medical Sciences, Beijing 100730, China; 6Department of Biostatistics, School of Public Health, University of Michigan, Ann Arbor, MI 48109, USA; kevinhe@umich.edu; 7Peking-Tsinghua Center for Life Sciences, Academy for Advanced Interdisciplinary Studies, Peking University, Beijing 100871, China; 8National Institute of Health Data Science at Peking University, Beijing 100191, China

**Keywords:** all-cause mortality, chronic kidney disease, genetic association, outcomes, single nucleotide polymorphism, *UMOD* gene

## Abstract

We aimed to explore associations of several single nucleotide polymorphisms (SNPs) detected by genome-wide association studies in *uromodulin* (*UMOD*) gene with phenotypes and prognosis of chronic kidney disease (CKD) among 2731 Chinese patients with CKD stage 1–4. Polymorphisms of rs11864909, rs4293393, rs6497476, and rs13333226 were genotyped using the Sequenom MassARRAY iPLEX platform. The SNPs of rs13333226 and rs4293393 were in complete linkage disequilibrium. Based on the T dominant model, T allele of rs11864909 was associated with levels of estimated glomerular filtration rate (eGFR) and serum uromodulin with linear regression coefficients of 2.68 (95% confidence interval (CI): 0.61, 4.96) and −12.95 (95% CI: −17.59, −7.98), respectively, after adjustment for cardiovascular and kidney-specific risk factors. After a median follow-up of 4.94 years, both G allele of rs4293393/rs13333226 and C allele of rs6497476 were associated with reduced risk of all-cause mortality with multivariable-adjusted hazard ratios of 0.341 (95% CI: 0.105, 0.679) and 0.344 (95% CI: 0.104, 0.671), respectively. However, no associations were found between the variants and slope of eGFR in the linear mix effect model. In summary, the variant of rs11864909 in the UMOD gene was associated with levels of eGFR and serum uromodulin, while those of rs4293393 and rs6497476 were associated with all-cause mortality among patients with CKD.

## 1. Introduction

Uromodulin, also known as Tamm-Horsfall protein, is the most abundant protein in human urine, which is exclusively synthesized in the thick ascending limb (TAL) of the loop of Henle in the kidney. The protein has important roles in ion transport, maintenance of water and electrolyte balance, protection against urinary tract infection, and kidney innate immunity [1,2]. Besides being excreted into the tubular fluid through an intracellular route to the apical cell pole, uromodulin can be released into plasma by transport to the basolateral cell site via the Golgi apparatus and cytoplasmic vesicles. As the measurement of urinary uromodulin may be influenced by specific preanalytic conditions, such as centrifugation, vortexing, and conditions and duration of storage, serum uromodulin may represent a more stable concentration of the biomarker [3]. A recently published prospective cohort study recruiting patients undergoing coronary angiography demonstrated a higher level of serum uromodulin was associated with an imporved metabolic profile and reduced risk of mortality after a median of 9.9 years of follow-up [4]. Consistently, our study team reported lower levels of serum uromodulin were independently associated with a higher risk of incident end-stage kidney disease (ESKD) among patients with chronic kidney disease (CKD) [5].

Mutations in the encoding gene of uromodulin, the *UMOD* gene, have been found to be associated with hereditary autosomal-dominant tubulointerstitial diseases [6]. In addition, several genome-wide association studies (GWAS) have identified common variants in the promoter region of the *UMOD* gene relating to estimated glomerular filtration rate (eGFR), risk of CKD, and urinary/serum uromodulin levels, highlighting the role of uromodulin in the pathophysiology of CKD [7,8,9]. However, most GWASs were conducted among participants of European or African American ancestry. The only exception is the Asian Genetic Epidemiology Network study, providing evidence for the East Asia population. In the study, rs11864909, replacing rs12917707, was identified as the most significant genetic variant of the association with CKD-related traits [10]. The single nucleotide polymorphisms (SNP) of rs12917707, reported by the previous GWASs, have an extremely low minor allele frequency among the East Asia population.

Examination of associations between *UMOD* polymorphisms and CKD-related phenotypes and prognosis of the disease has implications in understanding pathophysiology of the disease. As East Asians were not well represented in the previous GWAS studies, further evidence among a cohort of the Chinese population may inform consistency and/or difference of the association between ethnicities. Hence, we aimed to explore the association between the candidate SNP in the promoter region of *UMOD* identified in previous GWAS and traits of CKD among participants in the Chinese Cohort Study of CKD (C-STRIDE). We hypothesized that the risk alleles (T allele of rs11864909, G allele of rs13333226, G allele of rs4293393, and C allele of rs6497476) can contribute to the variety of levels of eGFR and serum uromodulin and may have an impact on the prognosis and complications of CKD.

## 2. Materials and Methods

### 2.1. Study Samples

C-STRIDE is an ongoing cohort study initiated in November 2011, which includes adult patients (18–74 years old) with CKD stage 1–4 in 39 clinical centers around China. The eGFR of patients should be between specific ranges according to different etiologies of CKD. For glomerulonephritis, eGFR should be ≥15 mL/min/1.73 m^2^. For diabetic nephropathy, eGFR should be either between 15 mL/min/1.73 m^2^ and 59 mL/min/1.73 m^2^ or ≥60 mL/min/1.73 m^2^ with 24-hour urinary protein ≥3.5 g or urinary albumin-to-creatinine ratio (UACR) ≥2000 mg/g or equivalent levels of other proteinuria measurements. For the etiology other than glomerulonephritis and diabetic nephropathy, eGFR should be between 15 mL/min/1.73 m^2^ and 59 mL/min/1.73 m^2^. The exclusion criteria included CKD caused by systemic inflammatory illness or autoimmune disease, isolated hematuria, hereditary kidney disease, kidney or other transplantation, treatment with immunosuppressive agents in the preceding 6 months to treat kidney or immune disease, HIV infection and/or diagnosis of AIDS, chronic heart failure with New York Heart Association Class III or IV, known diagnosis of cirrhosis, pregnancy or breast-feeding, malignancy treated with chemotherapy within last 2 years, and current participation in a clinical trial [11]. Totally, 3877 participants finished baseline examination between 1 November 2011 and 31 December 2017. Among the total population, 2754 participants had their blood samples shipped to the coordinating center. Of the 2754 participants, 2731 participants were successfully genotyped for the selected SNPs and included in the current analysis. The C-STRIDE study was conducted in accordance with the Declaration of Helsinki. The study has been approved by the Ethics Committee of Peking University First Hospital (approval number: 2011[363]). All participants provided informed consent.

### 2.2. Genotyping

Genomic DNAs of participants were isolated from blood leukocytes by the salting-out procedure. We genotyped 4 common variants in the promoter region of the *UMOD* gene (rs11864909, rs4293393, rs6497476, and rs13333226). Genotyping of the SNPs was carried out using the matrix-assisted laser desorption/ionization time-of-flight mass spectrometry (MALDI-TOF MS, Agena, SEQUENOM, Inc., San Diego, CA 92121, USA) according to the manufacturer’s instructions [12]. All the experiments were conducted by investigators who were blind to the phenotypes. Negative controls and duplicate samples were placed on each run to ensure correct genotyping. The genotyping call rates were 99.61% for rs11864909, 99.44% for rs4293393, 99.58% for rs6497476, and 99.58% for rs13333226.

### 2.3. Uromodulin Measurement

Uromodulin measurement was performed in the central laboratory of Peking University First Hospital, using fasting venous blood samples obtained at baseline study visit and stored at −80 °C until use. The detailed procedure for the measurement has been described in our previous publication [5]. Briefly, we measured serum uromodulin by a commercially available enzyme-linked immunosorbent assay kit (Euroimmun AG, Lübeck, Germany) according to the manufacturer’s instructions. At the mean concentration of 29.7 ng/mL, 102.0 ng/mL, and 214.4 ng/mL, the intra-assay coefficient of variation was 3.2%, 2.2% and 1.8%, respectively. The lower detection limit of the assay was 2.0 ng/mL.

### 2.4. Measurement of Covariates

The trained staff in each clinical center conducted the questionnaire and physical examinations. Similar to uromodulin, other serum and urinary biomarkers for the current study were measured centrally at the Peking University First Hospital. The measurements of serum and urine creatinine were traceable to the isotope dilution mass spectrometry. eGFR was determined using the CKD-EPI creatinine equation [13]. The UACR (mg/g creatinine) was calculated. The classification for the eGFR and ACR was determined according to the Kidney Disease Improving Global Outcomes guideline [14]. Twenty-four-hour urine was collected and urinary sodium excretion was measured. Body mass index was calculated as weight in kilograms divided by the square of height in meters (kg/m^2^). Blood pressure (BP) was measured three times at 5-minute intervals by a sphygmomanometer at the follow-up visit. The mean value of the three readings was calculated. Either abnormal BP (systolic BP ≥ 140 or diastolic BP ≥ 90) or using anti-hypertensive medications in the past two weeks was defined as hypertension. Diabetes was defined as either fasting blood glucose ≥7.0 mmol/L or a self-reported history of diabetes. History of cardiovascular disease (CVD) included a self-reported history or reviewing of medical records at baseline for myocardial infarction, serious cardiac arrhythmia, peripheral arterial disease, cerebrovascular events, or hospitalization for congestive heart failure.

### 2.5. Outcomes

#### 2.5.1. ESKD, CVD Events and All-Cause and CVD-Specific Mortality

ESKD, CVD events, and mortality were followed up until 31 December 2017. ESKD is defined as the initiation of hemodialysis, peritoneal dialysis, or kidney transplantation. CVD events include non-fatal acute myocardial infarction, unstable angina, hospitalization for congestive heart failure, arrhythmia (including resuscitated cardiac arrest, ventricular fibrillation, sustained ventricular tachycardia, paroxysmal ventricular tachycardia, an initial episode of atrial fibrillation or flutter, severe bradycardia or heart block), cerebrovascular events (including intraparenchymal hemorrhage, subarachnoid hemorrhage, and cerebral infarction), and peripheral vascular diseases. Outcomes were investigated at a three to six-month interval through phone calls or routine clinical visits. Medical records were used to verify suspected outcomes of ESKD and CVD. Causes of death in ICD-10 codes I00-I99 were classified as CVD. An independent committee consisting of specialist physicians in Peking University First Hospital adjudicated the outcomes. If several CVD events occurred, the first event was used as the index event. We censored CVD events at the occurrence of ESKD, death or end of follow-up (31 December 2017), ESKD at death or the end of follow-up, all-cause mortality at the end of follow-up, while CVD-specific mortality at death from other reasons or the end of follow-up.

#### 2.5.2. eGFR Slope

A subgroup of patients, who had repeated measures of eGFR (≥2 times) with the first and last measures spanning ≥1 year, were used to estimate eGFR slope, representing the rate of eGFR change. A sensitivity analysis was conducted for those with ≥3 times of repeated measures of eGFR. Consistently, the time period between the first and last measures should be ≥1 year.

#### 2.5.3. Statistical Analysis

Continuous data were presented as mean ± standard deviation or median (interquartile range), while categorical data were expressed as counts (percentage). One-way ANOVA was used to compare means of continuous variables conforming to a normal distribution, while the Kruskal–Wallis test was used in case of skewed distribution. A chi-square test was employed to compare proportions of categorical variables and to test Hardy–Weinberg equilibrium. Lewontin’s D’, the logarithm of the odds score, and r2 were calculated to estimate the correlation and magnitude of linkage disequilibrium (LD) between SNPs. In order to incorporate haplotypes spanning the studied SNPs into the regression analysis, we used the expectation-maximization algorithm based on the HAPLOTYPE procedure in SAS software (version 9.4) to generate maximum likelihood estimates of haplotype frequencies given the genotypes of the selected SNPs. Whereby, probabilities of possessing different haplotypes were assigned to each individual. The associations between genotypes of a single SNP or haplotype spanning multiple loci and concentrations of eGFR or serum uromodulin were examined by a general linear regression model. Variance inflation factors were calculated for the covariates included in the multivariable model to detect potential multi-collinearity. If significant associations were detected, we further conducted analysis stratified by groups of age, hypertension, eGFR, UACR, and etiologies of CKD to reflect the heterogeneity of the association. Incidence rates of ESKD, CVD events, all-cause, and CVD-specific mortality were calculated and compared through genotypes of SNPs by log-rank test. If significant differences were detected in the log-rank test, Cox proportional hazards regression model was used to quantify the association between genotype or haplotype and the outcomes. The proportional hazards assumption was tested by Schoenfeld residuals. As inspired by previous publications, demonstrating hypertension and urinary sodium excretion to be of potential effect modification on the association between the *UMOD* gene and the risk of mortality [15,16], we included interaction terms between the studied SNPs and hypertension or 24-hour urinary sodium excretion into the Cox regression model. In addition, in order to test the influence of the SNPs on the progression of CKD, a linear mixed-effects model was used to calculate eGFR slopes between the genotypes or haplotypes of the SNPs. Follow-up time, genetic variants, interaction term between follow-up time, and genetic variants were included as fixed effect items with an unstructured variance-covariance matrix, random intercept, and random follow-up time.

The bootstrap method with 500 times of sampling with replacement was conducted to generate a 95% confidence interval (CI) in the general linear regression and Cox regression model. As 95% CIs were very close between the bootstrap method and the theoretic method, we only presented results from the bootstrap method. *P* values less than 0.05 were considered statistically significant. LD was estimated by using Haploview software [17]. All other analyses were conducted by using SAS software (version 9.4, SAS Institute Inc, Cary, NC, USA).

## 3. Results

The flowchart of selecting study participants was shown in Figure 1. The participants had a mean age of 48.94 years with 59.83% of male and the majority of them were of Han ethnicity (93.00% vs. 3.24% of other ethnicities and 3.76% missing). The distribution of all four studied genetic variants complied with Hardy–Weinberg equilibrium (all *p*-values > 0.05). No significant differences exist for the distribution of the variants by comparing Han ethnicity with other ethnicities. The number of participants with CC, TC, and TT genotype of rs11864909 was 1,988, 676, and 67, respectively. The measurements for the correlation and magnitude of LD between the SNPs were listed in Appendix A. rs13333226 and rs4293393 were in complete LD, so only rs4293393 was used in the following analyses. Patients had lower levels of serum uromodulin, but higher levels of eGFR through the CC, TC to TT genotypes of rs11864909 (both *p*-values < 0.05). Although the A allele of rs4293393 and T allele of rs6497476 was shown to be associated with higher levels of both eGFR and serum uromodulin, the differences did not reach statistical significance due to the corresponding alleles accounting for the majority of the population (Figure 2). 

Because of the limited sample size of the TT genotype, we combined TC and TT genotypes of rs11864909 together and compared characteristics of patients between CC versus TC&TT genotypes. Besides higher levels of eGFR and lower levels of serum uromodulin, patients with TC&TT genotypes of rs11864909 had more male and more carriers of AA genotype of rs4293393 and TT genotype of rs6497476 (all *p*-values < 0.05) (Table 1). The distributions of covariates stratified by genotypes of rs13333226/rs4293393 or rs6497476 were shown in Appendix A. We also compared characteristics between those included in and excluded from the analysis. The level of eGFR was comparable between the populations. However, participants included in the analysis were younger, more likely to smoke and have a CVD history, less likely to have an etiology of glomerulonephritis, had a much higher level of UACR, but a lower level of systolic BP than those excluded (all *p*-values < 0.05, in Appendix A).

The level of eGFR and serum uromodulin was tightly correlated with the Pearson correlation coefficient of 0.67 (*p*-value < 0.001). The TC and TT genotypes of rs11864909 were associated with an increased level of eGFR, but a reduced level of serum uromodulin with regression coefficients of 2.68 (95% CI: 0.61, 4.96) and −12.95(95% CI: −17.59, −7.98), respectively, in the fully adjusted regression model. The haplotype composed of rs11864909, rs4293393, and rs6497476 (TAT vs. other types) was positively associated with eGFR but negatively associated with serum uromodulin with an augmented effect size than those with respect to the single SNP of rs11864909 (Table 2). No significant association was found for either rs4293393 or rs6497476 with the two serum phenotypes (Appendix A). Stratified analysis showed that the associations of rs11864909 or haplotype spanning rs11864909, rs4293393, and rs6497476 with either eGFR or serum uromodulin were stronger among patients aged < 65 years, without hypertension, with normal eGFR (≥60 mL/min/1.73 m^2^), A3 stage of UACR (≥300 mg/g), and etiology of glomerulonephritis (Appendix A). In addition, no indication of serious multiple collinearity was detected given variance inflation factors for all covariates were less than 5.

In the survival analysis, the GA and GG genotypes of rs4293393 and TC and TT genotypes of rs6497476 were combined together to make a balanced sample size between genotype groups. The follow-up time for ESKD, CVD events, and all-cause mortality were 4.68 (interquartile range: 3.87–5.59), 4.79 (4.07–5.92), and 4.94 (4.13–5.97) years, respectively. We found no significant difference for the incidence of ESKD, CVD events, and CVD-specific mortality between the genotype groups of all the studied SNPs (all *p*-values of log-rank test > 0.05). However, there was a significantly higher incidence of all-cause mortality among patients with the AA genotype of rs4293393 or TT genotype of rs6497476 (both *p*-values of log-rank test < 0.05) (Table 3). In the Cox regression analysis, GG&GA vs. AA genotype of rs4293393, CC&TC vs. TT genotype of rs6497476, and CGC vs. other phases of haplotype spanning rs11864909, rs4293393, and rs6497476 were associated with reduced risk of all-cause mortality in the multivariable-adjusted model, with HRs of 0.341 (95% CI: 0.105, 0.679), 0.344 (95% CI: 0.104, 0.671) and 0.118 (95% CI: 0.011, 0.446), respectively (Table 4). No violation of the proportional-hazards assumption was found for the genetic variants and covariates after assessment of the Schoenfeld residuals. The addition of serum uromodulin into the regression model with the other risk factors could enhance the magnitude of HRs. We detected a significant effect modification between 24-hour urinary sodium excretion and either genotypes of rs4293393 or those of rs6497476 (both *p*-values for interaction <0.01) in the fully adjusted model, but not any significant interactions were detected between hypertension and the genetic variants. Levels of 24-hour urinary sodium excretion through genotypes of rs4293393 or rs6497476 are contained in Appendix A. We stratified the association by the median level of 24-hour urinary sodium excretion (135 mmol/24 h). As there were very few cases of all-cause mortality occurred in the GA & GG genotypes of rs4293393 or in the CT & CC genotypes of rs6497476, making the regression analysis impossible, we only listed the incidence rates of the outcomes in the stratified analysis. The difference for the incidence of all-cause mortality was only prominent among those with higher levels of 24-hour urinary sodium excretion (≥135 mmol/24 h) (Appendix A).

Among the participants with longitudinal measurement of eGFR (two or more times lasting for more than one year, *n* = 1337), there are lower decline rates of eGFR among those with CC genotype of rs11864909, GA & GG genotypes of rs4293393, CT & CC genotypes of rs6497476 or haplotype phases of TAT spanning the SNPs. However, the differences of the eGFR slopes between the genotypes or haplotype phases were not statistically significant (all *p*-values for interaction between the genetic variants and time >0.05) (Appendix A). Sensitivity analysis included participants with three or more times of measurements of eGFR lasting for more than one year (*n* = 1046) yielded consistent results (Appendix A).

We made a comparison between main findings of our study findings and those reported by previous studies in Table 5.

## 4. Discussion

In this cohort study of patients with predialysis CKD of Chinese ethnicity, we found that the T allele of rs11864909 in the promotor of the *UMOD* gene in the dominant model was associated with higher levels of eGFR and lower levels of serum uromodulin. The associations were consistent through different age, hypertension, eGFR, urinary ACR, and causes of CKD groups. Furthermore, rs13333226, rs4293393, and rs6497476, also common variants of the *UMOD* gene, were associated with the occurrence of all-cause mortality.

Several common variants located in the *UMOD* gene promotor, such as rs12917707, rs13333226, rs6497476, and rs4293393, have been detected significantly associated with CKD-related traits in the recently published GWAS among populations of European descent [7,9]. Regarding east Asians, another polymorphism in the promoter region of *UMOD*, rs11864909, was reported as the most significant genome-wide association signal, taking the place of rs12917707, owing to a very low minor allele frequency (<0.01) of the SNP among the population [10]. In the current study of patients with CKD, we used a candidate gene strategy to select the above SNPs and confirmed the previous findings that the T allele of rs11864909 was associated with an elevated level of eGFR. Although we also detected either A allele of rs4293393/rs13333226 or T allele of rs6497476 was associated with a higher concentration of eGFR, the differences failed to reach a significant threshold due to extremely high frequency of the alleles (99.7% for both A allele of rs4293393/rs13333226 and T allele of rs6497476). Besides eGFR, we measured serum uromodulin in the current study, which was positively correlated with the level of eGFR in previous studies [19,20]. However, contrary to the positive association with eGFR, the T allele of rs11864909 was associated with a lower level of uromodulin. Similar findings have also been reported by Graciela and colleagues, where rs12917707 was associated with serum uromodulin based on a GWAS among participants of the Ludwigshafen Risk and Cardiovascular Health Study (2826 of the 3316 population with genotyping data). Likewise, genotypes of rs4293393, rs13333226, and rs6497476 were not significantly associated with the level of serum uromodulin in their study [4].

Besides CKD-related phenotypes, a recent GWAS has also reported the association between common variants of the *UMOD* gene and hypertension [15]. Another study by Gudbjartsson et al. additionally found that variants of *UMOD* were more strongly associated with CKD among older adults and those with multiple comorbidities, such as hypertension, diabetes, and CVD, highlighting the role of the risk variants involved in the mechanisms relating to adaptation to aging [9]. In fact, a basic medicine study, conducted by Trudu and colleagues, provided compelling evidence regarding the mechanisms linking the variants of *UMOD* and the development of hypertension and kidney lesions. They found overexpression of uromodulin due to the presence of *UMOD* risk variants both in vitro and in vivo, which could cause over-activation of TAL sodium–potassium–chloride co-transporter, leading to salt-sensitive hypertension and age-dependent kidney lesions. Blocking the pathway by diuretics could result in a drop in BP [16]. In the current study, we did not detect a significant relationship between all the studied SNPs and BP. However, we found that associations between rs11864909 and eGFR or serum uromodulin were more prominent among participants with higher urinary ACR (≥300 mg/g), which represents advanced kidney injury and is highly correlated with long-term hypertension and diabetes. However, we do not find advanced age or hypertension could modify the effect of the variant.

Regarding the relationship between variants of the *UMOD* gene and adverse outcomes of CKD, the above-mentioned Ludwigshafen Risk and Cardiovascular Health Study also reported that the T allele of rs12917707 was shown to be associated with reduced risk of mortality among those aged <67 years after a median 9.9 years of follow-up [4]. In our study, the majority of the population was also aged <67 years (89.53%) and we found consistent results that the G allele of rs4293393/ rs13333226 and C allele of rs6497476 in the dominant model were associated with reduced risk of mortality. There is evidence supporting the protective role of the serum uromodulin-lowering allele of the variants of the *UMOD* gene the regarding risk of CVD. Based on the large-scale Malmo Diet and Cancer study with the exclusion of prior CVD events at baseline and with a follow-up of 12 years (*n* = 26654), Sandosh et al. reported each copy of the G allele of rs13333226 was associated with a 7.7% reduction for risk of CVD after adjusting for age, sex, and body mass index. Only a tiny abbreviation was observed when SBP and/or DBP was added into the regression model [15]. Using an extreme case-control design with regard to levels of BP, the same authors also found a better cardiovascular risk profile associated with the G allele of rs13333226 [15]. However, except for relating to lower risk of all-cause mortality, the SNPs of rs4293393/rs13333226 and rs6497476 in the current study were not found to be associated with risk of other outcomes, such as non-fatal CVD events and CVD specific mortality. By comparing to the above-mentioned study of Sandosh and colleagues, our study may lack enough power to detect a moderate association due to a comparatively short term of follow-up and a well-managed population by nephrology specialties regarding the risk of CVD.

In our study, we also detected that the association between the G allele of rs4293393/ rs13333226 or the C allele of rs6497476 and reduced risk of mortality was only present in those with a larger than median level of 24 h-urinary sodium excretion (≥135 mmol/24 h). However, due to the quite limited number of events of mortality in the specific genotype groups, important confounding factors, ranging from demographics, comorbidities to eGFR or serum uromodulin, cannot be adjusted. Further studies involving a large sample size and with longer follow-up time as well as the inclusion of more events are needed to validate the findings.

The C-STRIDE study recruited a large sample of patients with CKD around mainland China and evaluated their clinical characteristics comprehensively. Despite the strengths of this study, there were some limitations. Baseline eGFR and serum uromodulin were measured only once and may be subject to instability. Follow-up time for progression of CKD and occurrence of outcomes was comparatively short, which may limit the study power due to the limited number of events recorded.

## 5. Conclusions

We report significant association from an East Asian-based GWAS for the relationship between rs11864909 in the promoter region of the *UMOD* gene with phenotypes of eGFR and serum uromodulin among patients with CKD of Chinese ethnicity. In addition, the other three variants in *UMOD*, rs4293393, rs13333226, and rs6497476, were associated with the risk of all-cause mortality in the population. The results provided further evidence regarding the pathogenesis and prognosis of CKD related to the *UMOD* gene.

## Figures and Tables

**Figure 1 genes-12-01687-f001:**
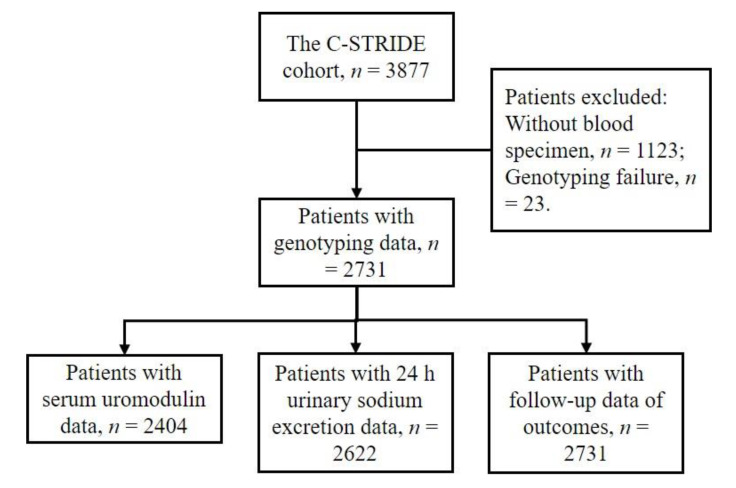
Flowchart of the participants selection.

**Figure 2 genes-12-01687-f002:**
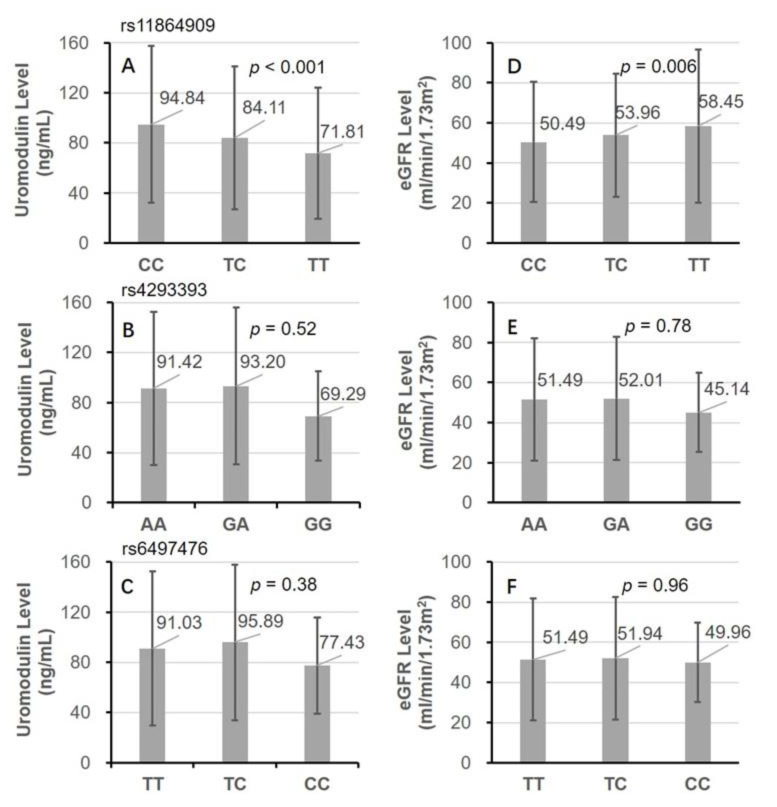
The distribution of serum uromodulin and eGFR between genotypes of rs11864909, rs4293393, and rs6497476. (**A**) Uromodulin levels through genotypes of rs11864909; (**B**) uromodulin levels through genotypes of rs4293393; (**C**) uromodulin levels through genotypes of rs64974763; (**D**) eGFR levels through genotypes of rs11864909; (**E**) eGFR levels through genotypes of rs4293393; (**F**) eGFR levels through genotypes of rs64974763.

**Table 1 genes-12-01687-t001:** Characteristics of study participants stratified by genotypes of rs11864909.

Characteristics	Total	C/C	T/C & T/T	*p*-Value
	*n* = 2731	*n* = 1988	*n* = 743	
Age, years	48.94 ± 13.81	49.13 ± 13.78	48.42 ± 13.88	0.23
Male, *n* (%)	1634 (59.83%)	1160 (58.35%)	474 (63.80%)	0.01
High school and above, *n* (%)	1509 (55.70%)	1090 (55.16%)	419 (57.16%)	0.35
Current and ever smoking, *n* (%)	1043 (39.40%)	737 (38.35%)	306 (42.21%)	0.07
Body mass index, kg/m^2^	24.50 ± 3.62	24.42 ± 3.57	24.71 ± 3.74	0.07
Systolic blood pressure, mmHg	129.58 ± 17.98	129.71 ± 17.74	129.25 ± 18.59	0.57
Diastolic blood pressure, mmHg	80.97 ± 10.95	81.08 ± 10.95	80.67 ± 10.94	0.42
Using anti-hypertensive medication, *n* (%)	1620 (73.01%)	1169 (73.06%)	451 (72.86%)	0.92
Diabetes mellitus, *n* (%)	625 (25.52%)	448 (25.13%)	177 (26.58%)	0.46
History of CVD, *n* (%)	277 (10.14%)	193 (9.71%)	84 (11.31%)	0.22
Creatinine, μmol/L	143 (100, 207)	145 (102, 210)	137 (93, 199)	0.02
eGFR, mL/min/1.73 m^2^	51.54 ± 30.44	50.49 ± 29.98	54.37 ± 31.49	0.003
eGFR <60mL/min/1.73 m^2^, *n* (%)	1858 (68.03%)	1381 (69.47%)	477 (64.20%)	0.009
ACR, mg/g	435.59 (114.00, 991.20)	434.85 (116.82, 985.43)	437.38 (108.20, 1018.61)	0.92
Albuminuria groups, *n*(%)				0.88
<30 mg/g	315 (11.81%)	233(11.99%)	82 (11.33%)	
30–299 mg/g	770 (28.86%)	562 (28.91%)	208 (28.73%)	
≥300 mg/g	1583 (59.33%)	1149 (59.10%)	434 (59.94%)	
Uromodulin, ng/mL	91.60 ± 61.37	94.84 ± 62.79	82.99 ± 56.57	<0.001
Etiology of CKD				0.63
Diabetic nephropathy	392 (14.77%)	286 (14.83%)	106 (14.60%)	
Glomerulonephritis	1605 (60.47%)	1156 (59.96%)	449 (61.85%)	
Others	657 (24.76%)	486 (25.21%)	171 (23.55%)	
Genotype of rs13333226				<0.001 ^1^
AA	2333 (85.43%)	1668 (83.90%)	665 (89.50%)	
GA	389 (14.24%)	313 (15.74%)	76 (10.23%)	
GG	9 (0.33%)	7 (0.35%)	2 (0.27%)	
Genotype of rs4293393				<0.001 ^1^
AA	2333 (85.43%)	1668 (83.90%)	665 (89.50%)	
GA	389 (14.24%)	313 (15.74%)	76 (10.23%)	
GG	9 (0.33%)	7 (0.35%)	2 (0.27%)	
Genotype of rs6497476				<0.001 ^1^
TT	2382 (87.22%)	1693 (85.16%)	689 (92.73%)	
TC	342 (12.52%)	288 (14.49%)	54 (7.27%)	
CC	7 (0.26%)	7 (0.35%)	0 (0.00%)	

Note: The *n* indicates number of subjects. The *p*-value indicates statistical significance for hypothesis testing. Number of missing: education-22, smoking status-84, body mass index-263, systolic blood pressure -379, diastolic blood pressure -379, using anti-hypertensive medication-512, diabetes mellitus-282, etiology of CKD-77, ACR-63, uromodulin-327. Abbreviation: CVD, cardiovascular disease; eGFR, estimated glomerular filtration rate; ACR, albumin creatinine ratio; CKD, chronic kidney disease. ^1^ Fisher’s exact test was used.

**Table 2 genes-12-01687-t002:** Genotypes of rs11864909 and haplotypes involving rs11864909, rs4293393 and rs6497476 and eGFR or uromodulin.

Association Model	β (95% CI) for eGFR	β (95% CI) for Uromodulin ^1^
rs11864909 (TT & TC vs. CC)		
Model 1	3.88 (1.25, 6.62)	−11.78 (−16.79, −6.12)
Model 2	3.36 (0.94, 6.03)	−12.10 (−17.27, −6.68)
Model 3	2.68 (0.61, 4.96)	−12.95 (−17.59, −7.98)
Model 4	5.65(4.02, 7.67)	−18.24 (−22.27, −14.12)
Haplotype composed of rs11864909, rs4293393 and rs6497476 (TAT vs. other types)		
Model 1	7.85 (3.17, 13.31)	−19.78 (−28.46, −9.80)
Model 2	6.80 (2.74, 12.17)	−20.62 (−29.38, −11.09)
Model 3	5.71 (2.01, 10.12)	−22.22 (−29.87, −13.20)
Model 4	10.90 (7.63, 14.77)	−32.73 (−39.52, −25.44)

Note: The results listed were gained by using the bootstrap method after 500 times of sampling with replacement. Model 1 was unadjusted. Model 2 was adjusted for age and gender. Model 3 was additionally adjusted for smoking, body mass index, systolic blood pressure, using anti-hypertensive medication, diabetes mellitus, etiology of CKD, and logarithm transformed urinary albumin creatinine ratio. Model 4 was additionally adjusted for uromodulin or eGFR, as appropriate. Abbreviation: eGFR, estimated glomerular filtration rate. ^1^ There are 379 missing values for uromodulin.

**Table 3 genes-12-01687-t003:** Incidence rates for outcomes among all study participants and stratified by genotypes of variants.

Genetic Variant	ESKD			CVD Events			All-Cause Mortality			CVD Specific Mortality		
	No. of Events (%)	Rate/100 Patient-Years	*p*-Value for Log-Rank Test	No. of Events (%)	Rate/100 Patient-Years	*p*-Value for Log-Rank Test	No. of Events (%)	Rate/100 Patient-Years	*p*-Value for Log-Rank Test	No. of Events (%)	Rate/100 Patient-Years	*p*-Value for Log-Rank Test
All patients	444 (16.26%)	3.66		218 (7.98%)	1.70		122 (4.47%)	0.91		48 (1.76%)	0.36	
rs11864909			0.97			0.74			0.50			0.34
TT&TC	121 (16.29%)	3.67		57 (7.67%)	1.63		30 (4.04%)	0.82		13 (1.75%)	0.36	
CC	323 (16.25%)	3.66		161 (8.1%)	1.72		92 (4.63%)	0.95		35 (1.76%)	0.36	
rs4293393			0.68			0.57			0.01			0.67
GA&GG	68 (17.09%)	3.82		29 (7.29%)	1.53		8 (2.01%)	0.41		4 (1.01%)	0.20	
AA	376 (16.12%)	3.63		189 (8.1%)	1.73		114 (4.89%)	1.00		44 (1.89%)	0.39	
rs6497476			0.84			0.31			0.02			0.90
CT&CC	56 (16.05%)	3.56		23 (6.59%)	1.39		7 (2.01%)	0.41		3 (0.86%)	0.17	
TT	388 (16.29%)	3.67		195 (8.19%)	1.75		115 (4.83%)	0.99		45 (1.89%)	0.39	

Abbreviation: ESKD, end-stage kidney disease; CVD, cardiovascular disease.

**Table 4 genes-12-01687-t004:** Association between genotypes of rs4293393 and rs6497476 and haplotypes involving the variants and all-cause mortality.

Association Model	Hazard Ratio (95% CI)
rs4293393 (GG & GA vs. AA)	
Model 1	0.388 (0.141, 0.752)
Model 2	0.399 (0.143, 0.772)
Model 3	0.370 (0.131, 0.697)
Model 4	0.341 (0.105, 0.679)
rs6497476 (CC & TC vs. TT)	
Model 1	0.410 (0.156, 0.748)
Model 2	0.420 (0.159, 0.755)
Model 3	0.375 (0.147, 0.699)
Model 4	0.344 (0.104, 0.671)
Haplotype composed of rs11864909, rs4293393 and rs6497476 (CGC vs. other types)	
Model 1	0.168 (0.025, 0.548)
Model 2	0.176 (0.026, 0.556)
Model 3	0.141 (0.022, 0.474)
Model 4	0.118 (0.011, 0.446)

Note: The results listed were gained by using the bootstrap method after 500 times of sampling with replacement. Model 1 was unadjusted. Model 2 was adjusted for age and gender. Model 3 was additionally adjusted for smoking, body mass index, systolic blood pressure, using anti-hypertensive medication, diabetes mellitus, etiology of chronic kidney disease, logarithm transformed urinary albumin creatinine ratio, and estimated glomerular filtration rate. Model 4 was additionally adjusted for uromodulin.

**Table 5 genes-12-01687-t005:** Single nucleotide polymorphisms in the promoter region of *UMOD* gene and their association with clinical phenotype and/or outcome of statistical significance in different studies.

Study			rs12917707 (G > T)		rs11864909 (C > T)		rs13333226 (A > G)		rs4293393 (A > G)		rs6497476 (T >C)	
Reference	Study Type	Population	MAF	Phenotype/Outcome	MAF	Phenotype/Outcome	MAF	Phenotype/Outcome	MAF	Phenotype/Outcome	MAF	Phenotype/Outcome
Anna Kottgen, 2009 [7]	GWAS	Europeans	0.18	CKD and eGFR	NA	NA	NA	NA	NA	NA	NA	NA
Daniel F. Gudbjartsson, 2010 [9]	GWAS	Icelanders	NA	NA	NA	NA	NA	NA	0.168–0.202	CKD and serum creatinine	NA	NA
Sandosh Padmanabhan, 2010 [15]	GWAS	Europeans	NA	NA	NA	NA	0.16-0.23	Hypertension, cardiovascular events and urinary uromodulin	NA	NA	NA	NA
Yukinori Okada, 2012 [10]	GWAS	East Asians	NA	NA	0.19	Serum creatinine and eGFR	NA	NA	NA	NA	NA	NA
Graciela E. Delgado, 2017 [4]	Candidate gene strategy	Europeans	0.19	eGFR and serum uromodulin; All-cause mortality (only among participants aged > 67 years)	NA	NA	NA	NA	NA	NA	NA	NA
Jia Han, 2013 [18]	Candidate gene strategy	Chinese	NA	NA	NA	NA	AA: 85.4%, GA + GG: 14.6%	Urine uric acid excretion and plasma uric acid	TT: 85.3%, CT + CC: 14.7%	Urine uric acid excretion and plasma uric acid	TT: 87.3%, CT + CC: 12.7%	Urine uric acid excretion and plasma uric acid
Jinwei Wang, current study	Candidate gene strategy	Chinese	NA	NA	0.148	eGFR and serum uromodulin	0.0745	All-cause mortality	0.0745	All-cause mortality	0.0652	All-cause mortality

Abbreviation: MAF, minor allele frequency; GWAS, genome-wide association study; NA, not-available; CKD, chronic kidney disease; eGFR, estimated glomerular filtration rate.

## Data Availability

The data presented in this study are available on request from the corresponding author. The data are not publicly available due to the restriction for raw data availability out of the participating study centers.

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
