# Peer review of "UMOD Polymorphisms Associated with Kidney Function, Serum Uromodulin and Risk of Mortality among Patients with Chronic Kidney Disease, Results from the C-STRIDE Study"

_genes, 2021, doi:10.3390/genes12111687_

Round 1
Reviewer 1 Report
I enjoyed reading this article, which is highly informative about UMOD snps in CKD. I appreciate all the work that such a longitudinal study entails, and I also know the statistical analysis is intricate. Congrats to the authors on putting this together.
I had several suggestions:
1) Could the authors state the primary hypothesis in the introduction and then address it clearly in the results.
2) The article provides an excellent analysis of the UMOD snps, but it is a little difficult to read through due to the many factors. Could the authors provide a table to guide us through? In the table, i would like to see the following: 1) Snps studied. 2) allele frequencies of the snps in the Asian and European populations from large population studies. 3) allele frequencies for each snp in the present study. 3) what are the known effects of each snp on urinary umod production and serum umod? 4) what are the major outcomes for the study for each snp?
3) The rs4293393 snp decreases umod production and is protective from development of chronic kidney disease. On the other hand low serum umod concentrations are associated with decreased gfr, likely correlating with kidney mass. Given this, could the authors look at the association between plasma umod and the various snps after adjusting for gfr? Could the authors discuss this a little as well?
Author Response
We really appreciate your praise for our work and the valuable comments. We will try to make our work better through making further revisions following your suggestions. The responses are as follows:
1) Could the authors state the primary hypothesis in the introduction and then address it clearly in the results.
- OK. We have included the primary hypothesis of the study in the last paragraph of the Introduction section. Hopefully, the hypothesis can help readers clearly expect what we would state in the Results section. The revision is as follows:
“Hence, we aimed to explore the association between the candidate SNP in promoter region of UMOD identified in previous GWAS and traits of CKD among participants in the Chinese Cohort Study of CKD (C-STRIDE). We hypothesized that the risk alleles (T allele of rs11864909, G allele of rs13333226, G allele of rs4293393 and C allele of rs6497476) can contribute to the variety of levels of eGFR and serum uromodulin and may have impact on the prognosis and complications of CKD.” in line 71-77.
2) The article provides an excellent analysis of the UMOD snps, but it is a little difficult to read through due to the many factors. Could the authors provide a table to guide us through? In the table, i would like to see the following: 1) Snps studied. 2) allele frequencies of the snps in the Asian and European populations from large population studies. 3) allele frequencies for each snp in the present study. 3) what are the known effects of each snp on urinary umod production and serum umod? 4) what are the major outcomes for the study for each snp?
- OK. We have included a table (Table 5) in the revised manuscript to make a comparison between our study findings and those reported by some major studies about the topic.
3) The rs4293393 snp decreases umod production and is protective from development of chronic kidney disease. On the other hand low serum umod concentrations are associated with decreased gfr, likely correlating with kidney mass. Given this, could the authors look at the association between plasma umod and the various snps after adjusting for gfr? Could the authors discuss this a little as well?
- Thank you for the comment. It is very interesting to observe that the loci in UMOD gene was associated with increased eGFR but reduced level of serum uromodulin, although eGFR and serum uromodulin were positively correlated in our study. The reason can lie on what you said. We have presented the results in Table 2(due to a non-significant association in crude analysis for rs4293393, we only presented the results regarding rs11864909, which has been found as an important loci among East Asian populations), where we adjusted eGFR and serum uromodulin mutually in model 4. We discussed the findings in the second paragraph of the Discussion section as follows:
“In the current study of patients with CKD, we used a candidate gene strategy to select the above SNPs and confirmed the previous findings that T allele of rs11864909 was associated with elevated level of eGFR. Although we also detected either A allele of rs4293393/rs13333226 or T allele of rs6497476 was associated with higher concentration of eGFR, the differences failed to reach significant threshold due to extremely high frequency of the alleles (99.7% for both A allele of rs4293393/ rs13333226 and T allele of rs6497476). Besides eGFR, we measured serum uromodulin in the current study, which was positively correlated with the level of eGFR in previous studies[19-20]. However, on the contrary to the positive association with eGFR, T allele of rs11864909 was associated with lower level of uromodulin. Similar findings have also been reported by Graciela and colleagues, where rs12917707 was associated with serum uromodulin based on a GWAS among participants of the Ludwigshafen Risk and Cardiovascular Health Study (2826 of the 3316 population with genotyping data). Likewise, genotypes of rs4293393, rs13333226 and rs6497476 were not significantly associated with the level of serum uromodulin in their study[4].” In line 306-320.
Reviewer 2 Report
This is an interesting study about the relevance of some UMOD gene polymorphisms as a risk factor for the development of CKD, which is truly one of the post prevalent chronic conditions in adult patients. I only have some minor comments/corrections:
- Line 63: “In the study, rs11864909, replacing rs12917707, which has a low minor allele frequency among population of east Asia, was identified as the most significant genetic variant of the association with CKD-related traits”. I suggest rephrasing this sentence as it is difficult to understand which of the 2 variants (rs11864909 or rs12917707) have a low minor allele frequency and is most significant.
- Table 3: difficult to read and understand because numbers are in different lines
- -Line 328: “In the current study, we did not detect a significant relationship between all the studied SNPs and BP, which was consistent with the findings by Gudbjartsson and colleagues”. I understand that the study by Gudbjartsson, mentioned in line 318, states the opposite, that is, “they found that variants at UMOD was more strongly associated with CKD among older adults and those with multiple comorbidities, such as hypertension, diabetes and CVD…”
Author Response
Thank you for your approval. Our revisions according to your suggestions are as follows:
1) Line 63: “In the study, rs11864909, replacing rs12917707, which has a low minor allele frequency among population of east Asia, was identified as the most significant genetic variant of the association with CKD-related traits”. I suggest rephrasing this sentence as it is difficult to understand which of the 2 variants (rs11864909 or rs12917707) have a low minor allele frequency and is most significant.
- Thank you for the comment. We did not state it clearly enough. The sentence has been rephrased as “In the study, rs11864909, replacing rs12917707, was identified as the most significant genetic variant of the association with CKD-related traits[10]. The SNP of rs12917707, reported by the previous GWASs, has an extremely low minor allele frequency among population of east Asia.”.
2) Table 3: difficult to read and understand because numbers are in different lines
- We are sorry for not arranging the table properly. In the revised manuscript, we adjusted the paper orientation to make all numbers in each cell of the table in the same line.
3) Line 328: “In the current study, we did not detect a significant relationship between all the studied SNPs and BP, which was consistent with the findings by Gudbjartsson and colleagues”. I understand that the study by Gudbjartsson, mentioned in line 318, states the opposite, that is, “they found that variants at UMOD was more strongly associated with CKD among older adults and those with multiple comorbidities, such as hypertension, diabetes and CVD…”
- We agree with your comment. In the study by Gudbjartsson and colleagues, presence of hypertension and other comorbid conditions of aging could enhance the association between the T allele of rs4293393 and either SCr or CKD. The authors demonstrated the effect modification of aging represented by the number of comorbid conditions rather than straightly testing the relationship of the SNP and other risk factors. We have deleted the sentence “which was consistent with the findings by Gudbjartsson and colleagues”.